# The relation between home numeracy practices and a variety of math skills in elementary school children

**Cléa Girard****\*, Thomas Bastelica, Jessica Léone, Justine Epinat-Duclos, Léa Longo, Jérôme Prado\***

Lyon Neuroscience Research Center (CRNL), Experiential Neuroscience and Mental Training Team, INSERM U1028—CNRS UMR5292, University of Lyon, Lyon, France

\* clea.girard@etu.univ-lyon1.fr (CG); jerome.prado@univ-lyon1.fr (JP)

**Data Availability Statement:** The data underlying this study are publicly available at https://osf.io/vty3c/files/.

## Abstract

A growing number of studies suggest that the frequency of numeracy experiences that parents provide at home may relate to children's mathematical development. However, the relation between home numeracy practices and children's numerical skills is complex and might depend upon both the type and difficulty of activities, as well as the type of math skills. Studies have also argued that this relation may be driven by factors that are not systematically controlled for in the literature, including socio-economic status (SES), parental math skills and children's IQ. Finally, as most prior studies have focused on preschoolers, it remains unclear to what extent there remains a relation between the home numeracy environment and math skills when children are in elementary school. In the present study, we tested an extensive range of math skills in 66 8-year-olds, including non-symbolic quantity processing, symbolic number understanding, transcoding, counting, and mental arithmetic. We also asked parents to complete a questionnaire about their SES, academic expectations, academic attitudes, and the numeracy practices that they provide at home. Finally, we measured their arithmetic fluency as a proxy for parental math skills. Over and above differences in socio-economic status, parental arithmetic fluency, child's IQ, and time spent with the child, we found a positive relation between the frequency of formal numeracy practices that were at or above grade level and two separate measures of mental arithmetic. We further found that the frequency of these advanced formal numeracy practices was related to parents' academic expectations. Therefore, our study shows that home numeracy experiences predict arithmetic skills in elementary school children, but only when those activities are formal and sufficiently challenging for children.

## Introduction

Numeracy is fundamental to many aspects of professional and personal life in modern society [1]. Yet, there are substantial individual differences in math skills among children [2]. A growing body of research indicates that these individual differences can be observed even before the beginning of formal schooling [3–6]. This raises the possibility that disparities in numerical

**Funding:** This research was supported by grants from the Agence Nationale de la Recherche (ANR-14-CE30-0002 and ANR-17-CE28-0014) to JP.

**Competing interests:** The authors have declared that no competing interests exist.

skills among children might (at least in part) come from their experiences with math at home [7–12].

The idea that children's home numeracy experiences may provide a foundation for math learning in school is motivated by decades of research showing that children's early literacy activities at home (e.g., reading and conversing with their caregivers) relate to future literacy skills [13–17]. Thus, in the same way as early home literacy activities define what has been termed the "home literacy environment" (HLE), activities and resources that relate to numeracy learning at home may define a "home numeracy environment" (HNE) [7–10]. Although research on the HNE remains more limited than research on the HLE, a growing number of studies for the most part relying on parental questionnaires [18–20], suggests associations between home numeracy practices and children's math outcomes [21–24]. However, the relation between math skills and the HNE appears to be relatively complex and sometimes inconsistent (for a review, see [25, 26]). For example, several studies have found a relation between higher quality HNE (i.e., more frequent home numeracy activities) and better numerical skills (e.g., [8, 10, 19, 20, 22, 24, 27–32]). However, other studies have failed to find such an association (e.g., [33–36]) and some have even found a negative relation [22, 37]. How can this relative inconsistency be explained?

## The HNE may involve practices that differ in type and difficulty

As a first possible way to explain the inconsistent relation between the HNE and math skills, LeFevre and colleagues have argued that home numeracy activities may have different effects depending on whether they are formal and informal [20, 22, 38]. Both formal and informal activities involve manipulating math concepts. However, formal activities are those in which parents explicitly intend to teach numerical skills to their child. For instance, parents clearly intend to teach math concepts when they engage in counting activities or explain the meaning of Arabic numerals. In contrast, informal activities are those whose purpose is not to teach numerical skills. Rather, these activities represent informal opportunities for children to be exposed to various math concepts, such as fraction when cooking or numerical ordering when playing a linear board game [20, 22]. Consistent with a conceptual dissociation between formal and informal numeracy activities, a few studies have found that the relation between the HNE and math skills differs as a function of the type of activity [20, 22, 39–41]. Note, however, that the pattern of differences is not necessarily consistent between samples, with some studies reporting a unique relation between formal numeracy practices and math skills [19, 39] and others reporting a unique relation between informal numeracy practices and math skills [20].

It is also possible that the inconsistent relation between the HNE and math skills may be explained by differences in the level of complexity of home activities. Skwarchuk [42] proposed that a distinction should be made between (a) formal activities that are basic and typically do not challenge children (e.g., counting a small number of objects in preschoolers) and (b) formal activities that are more advanced and typically do challenge them (e.g., count by twos in preschoolers). Overall, this conceptual distinction has been broadly supported in the literature. For instance, studies in which basic and advanced formal activities are distinguished have found a positive relation between children's math skills and the frequency of advanced formal numeracy activities [22, 23, 42, 43]. In contrast, the frequency of basic numeracy activities has been found to be either negatively correlated or not associated with children's math skills [44]. Overall, then, both differences in type (formal versus informal) and difficulty (basic versus advanced) of numeracy activities need to be considered when investigating a potential link between the HNE and math skills.

## The HNE may relate to some math skills and not others

The relation between the HNE and math skills may not only vary with the heterogeneity of home numeracy practices, but also with the type of children's math skills. In other words, the HNE might relate to some aspects of mathematical skills but not others. Although the large majority of studies on the HNE have used composite math scores (e.g., [10, 20, 30, 31, 35, 43, 45, 46]), there is growing evidence that the relation between the HNE and children's math skills may depend upon the type of skill measured [22, 24, 41, 47]. For example, Skwarchuk and colleagues [22] found that the frequency of formal home numeracy practices relates to symbolic number knowledge, but not to non-symbolic arithmetic skills. In contrast, the frequency of informal home numeracy practices relates to non-symbolic arithmetic skills, but not to symbolic number knowledge. More recent studies also found that the relation between the HNE and children's math skills depends upon the type of math ability (e.g., symbolic number processing, mapping, non-symbolic number processing, and calculation) as well as the type of numeracy practice (i.e., formal versus informal) [24, 41]. Therefore, in addition to differences in the type and difficulty of numeracy activities, differences in the type of math skills investigated may be an important factor when investigating the HNE [25]. Yet, many studies on the HNE lack a comprehensive assessment of a range of numerical skills in children [41].

## HNE and math skills in elementary school

Another dimension that might affect the relation between the HNE and math skills is the age of children. To date, the large majority of studies have examined the link between HNE and math performance in pre-schoolers and kindergarten-aged children [19, 20, 22, 40, 42]. Furthermore, the few studies that have focused on the HNE of older children do not allow for a detailed examination of the relations between numeracy practices and math achievement. For instance, Tamis-LeMonda and colleagues [11] have examined the relation between learning environment and math problem-solving skills in 5th-graders. Yet, quality of the environment was exclusively defined by literacy activities and general mother-child interactions. Sammons and colleagues [48] also related the frequency of educational practices at home to students' math achievement in secondary school. However, numeracy practices were not examined independently from literacy practices, making it impossible to distinguish the specific effects of the HNE. Finally, Lefevre and colleagues [20] gathered a more extensive measure of numeracy practices in parents of 2nd-graders. However, these children were grouped with kindergarten-aged and 1st-graders in the analyses. This again makes it difficult to assess the relation in older children (for a similar conclusion, see [25]). Therefore, little is known about the relation between math skills and the HNE beyond 1st grade (i.e., when children learn formal math in school). Yet, studies on academic socialization of children within the family context suggest that parental involvement in children's academic achievement continues well into elementary school (before it may decrease as children move to middle and high school; [49, 50]). For example, parents of elementary school children may vary in the extent to which they volunteer at school, attend school events, attend to meetings with teachers or engage in learning activities at home [51]. Such behaviors and beliefs towards schooling may be influenced by parents' own experiences [49] and shape children's academic development. Indeed, studies have found a relationship between parents' involvement in children's schooling and academic performance [52].

## The relation between HNE and math skills might be confounded by other environmental and genetic factors

Finally, previous studies have not always examined the relation between home numeracy practices and math skills while controlling from other potentially confounding factors. For

example, the development of numeracy skills is related to the development of literacy skills [53–57]. Although several studies have measured the influence of the home environment through general measures of both literacy and numeracy (i.e., the Home Learning Environment; [9, 58, 59]), little research has investigated the influence of both literacy and numeracy practices on children's math skills. Results from these studies are also mixed [24, 32, 35, 60]. For example, Anders and colleagues [27] found that measures of both home numeracy and literacy practices are related to young children's numerical development. Lehrl and colleagues [60] also found that book exposure during the preschool years predicted math skills later on. However, other studies showed more domain-specific literacy and numeracy practices [39, 61] or failed to show an effect of the home literacy practices on children's numerical skills [19, 35]. Therefore, it remains unclear whether children's numeracy skills specifically relate to home numeracy practices or depend more generally on the quality of the broad learning environment at home.

It is also important to note that the relation between quality of the home learning environment and academic skills is often interpreted as reflecting a purely environmental influence on children's abilities. However, this association might also be due (at least in part) to genetic heritability [62, 63]. Genetic predispositions to high math or reading achievement from parents may be passed on to children. These parents may also be more likely to maintain high quality learning environments [29, 64, 65]. In the case of literacy skills, van Bergen and colleagues [63] have recently shown that many (but not all) associations between home literacy practices and children's reading fluency were accounted for by parental reading fluency. This suggests that the relation between the HLE and literacy skills may reflect a mixture of environmental and genetic effects. Hart and colleagues have also argued that relations between HNE and children's math skills should be controlled for parental math skills, as it may serve as a genetic proxy for predisposition to high math achievement [62, 66]. To date, only a limited number of studies have included parents' math skills when studying the relation between the HNE and children's math achievement [29, 62, 67–69]. Therefore, it is possible that this relation reflects (at least in part) a passive gene–environment correlation.

## The current study

The present study investigates the relation between home numeracy practices and math skills of a sample of French early elementary schoolers (2nd and 3rd grades) who underwent more comprehensive behavioral testing than what is usually achieved in the HNE literature [10, 30, 41, 46]. Specifically, children were tested on a range of standardized math competences (non-symbolic quantity processing, symbolic number understanding, transcoding, counting, arithmetic calculation, and arithmetic fluency), as well as on measures of general cognitive functioning. Parents were also tested on a proxy measure of math skills (i.e., arithmetic fluency) and reported the frequency of informal, formal basic and formal advanced home numeracy (and literacy) practices.

This extensive data collection allowed us to address two main questions. First, we asked whether there remains a relation between home numeracy practices and math skills in these children, who are older than children typically tested in the HNE literature. Critically, our range of measures allowed us to (a) control for parental math skills, (b) assess which numeracy practices (informal versus formal, basic versus advanced) relate to specific math skills, and (c) evaluate whether the relation would be specific to numeracy practices (i.e., not observed with comparable literacy practices). In keeping with the previous literature on younger children, we expected to observe a relation between the HNE and math skills that would depend upon the type and difficulty of the practice. That is, we anticipated that informal practices would be

associated with non-symbolic skills while formal practices would be associated with symbolic skills, especially when those were relatively advanced with respect to children's age.

Second, we also aimed to investigate which parental traits (e.g., expectations, attitudes, skills) predict more frequent numeracy practices. Indeed, several studies have investigated the role of parental math attitude and academic expectations in shaping the HNE [see 25, 26]. Parental attitudes include their level of math anxiety (i.e., their tendency to avoid situations that involve math [70]) but also their use of math in everyday life as well as their evaluation of their own math skills when they were in school [71, 72]. Academic expectations are typically measured by asking parents how important it is for them that their child acquires a given skill within some time frame (e.g., [22]). Overall, although an association between parental attitudes and home numeracy practices remains to be clearly demonstrated (e.g., [20, 22]), studies have more consistently found a relation between higher academic expectations and increased frequency of numeracy practices at home [18, 19, 22, 30, 43]. Thus, we expected that the frequency of math activities reported by parents would depend on their academic expectations.

## Material and methods

### Participants

Seventy-three elementary school children of approximately 8 years of age and one of their parents were recruited for the experiment in the Lyon area in France. Participants were involved in a large project involving MRI scanning, which is why they all underwent extensive behavioral testing (at least as compared to what is typically achieved in studies on the HNE [25, 26]). All participants were contacted through flyers sent to schools and advertisements on social media. All parents and children came to the lab for a 2-hour long session during which they were given tests and questionnaires. Because studies suggest that the home learning environment may differ in children with low intelligence quotient (IQ) [73] and attention deficit disorder [74], we excluded children who had an IQ lower than the 25th percentile (n = 2) and were diagnosed with attention deficit disorder (n = 1). Because we also anticipated differences in the learning environment of children who experienced and were followed for language difficulties [75], we also excluded children who were seeing a speech-language pathologist on a regular basis (n = 3) and had a delay in speech and language acquisition (n = 1).

Therefore, 66 typically-developing 8-year-olds (M = 8.46, min = 7.51, max = 9.22) with no history of neurological disease, mental disorders of attention deficits were included in the final analyses. Out of the 66 children, 20 were in 2nd grade and 46 were in 3rd grade. A meta-analysis of the relations between children's math skills and predictors such as home numeracy practices, parental attitudes, and academic expectations estimated that the size of the correlation between correlations range from r = 0.31 to r = 0.46, with the smallest effect for academic expectations and the largest effect for practices (see Table 7.4 in [21]). Considering the smallest estimate of that range (r = 0.31), G*Power 3.1 [76] indicated that our final sample (n = 66) would provide an achieved power of 84% to detect a similar positive association in a linear regression at α = 0.05 (one tailed).

All children and parents included in the study were native French speakers. Children's full-scale IQ was normal to high-normal, ranging from 83 to 135 (average = 112, Standard Deviation [SD] = 11.8). Eighty-nine percent of the parents whose children were included in the analyses were mothers. Parents completed a questionnaire evaluating their SES. Parental income ranged from less than €12,000 to more than €60,000 per year. Fourteen percent of parents reported to only have a secondary degree, 50% reported to have an undergraduate degree, and 36% a master degree or higher. One parent did not go to high school. Therefore, SES ranged from low to high. Parents gave written informed consent and children gave their assent to

participate in the study. The study was approved by the local ethics committee (Comité pour la Protection des Personnes Sud-Est II, n°2014-041- AM2-2). Families were paid 40 euros for their participation in the testing session.

## Materials

**Child measures.**   Children's math skills were mainly assessed using the Neuropsychological Test Battery for Number Processing and Calculation in Children (Zareki-R, [77]). The battery includes 13 core subtests (see **S1 Appendix**). To reduce the number of variables, we grouped these subtests according to the corresponding math skill. Quantity estimation was measured using the "numerosity estimation" and "quantity in context" subtests. Symbolic number understanding was assessed using the "written comparison", "oral comparison", and "number-to-position mapping" subtests. Counting was assessed using the "dot counting" and "oral reverse counting" subtests. Transcoding was assessed using the "number reading" and "number writing" subtests. Finally, arithmetic calculation was assessed using the "oral problem-solving", "addition", "subtraction", and "multiplication" subtests. A standardized score was obtained for each of these subtests and an average standardized score was obtained for each math skill (see Table 1). Norms were based on a sample of 249 French children from 6 to 11.5 (i.e., Grade 1 to 5) [77]. In the norming sample, the Zareki-R battery has been shown to

**Table 1. Children's standardized (mean = 100, SD = 15) scores on the Zareki-R and WJ-III subtests.**

| Subtest | Mean (SD) | Min | Max |
|---|---|---|---|
| **Quantity estimation (Zareki-R)** | | | |
| **Numerosity estimation** | 106 (19) | 57 | 123 |
| **Quantity in context** | 109 (13) | 75 | 122 |
| **Average** | 107 (12) | 66 | 123 |
| **Symbolic number understanding (Zareki-R)** | | | |
| **Written comparison** | 106 (10) | 66 | 112 |
| **Oral comparison** | 110 (12) | 66 | 122 |
| **Number-to-position mapping** | 107 (11) | 73 | 129 |
| **Average** | 108 (7) | 90 | 121 |
| **Counting (Zareki-R)** | | | |
| **Dot counting** | 103 (11) | 58 | 111 |
| **Oral reverse counting** | 104 (2) | 68 | 114 |
| **Average** | 104 (7) | 80 | 113 |
| **Transcoding (Zareki-R)** | | | |
| **Number writing** | 107 (2) | 52 | 117 |
| **Number reading** | 105 (8) | 65 | 115 |
| **Average** | 106 (8) | 59 | 116 |
| **Arithmetic calculation (Zareki-R)** | | | |
| **Oral problem-solving** | 110 (13) | 84 | 127 |
| **Addition** | 107 (14) | 54 | 121 |
| **Subtraction** | 111 (12) | 83 | 128 |
| **Multiplication** | 107 (12) | 72 | 114 |
| **Average** | 109 (9) | 87 | 122 |
| **Arithmetic fluency (WJ-III)** | | | |
| **Math Fluency** | 104 (24) | 45 | 155 |

Notes. N = 66 The type of skill measured by each test is indicated in italics.

predict both teachers' evaluations of student's math skills, as well as to children's academic achievement in math [77]. A high internal consistency has also been reported in European samples ($\alpha$ = .93 to $\alpha$ = .97) [78].

In addition to the Zareki, children were also tested on the Math Fluency subtest of the Woodcock-Johnson Test of achievement (WJ III; [79]). In this test, participants solve simple addition, subtraction, and multiplication problems within a 3-min time limit. The test consists of 2 pages of 80 problems involving operands from 0 to 10. Addition, subtraction, and multiplication problems are intermixed, but multiplication problems are only introduced after Item 60. Because this subtest is timed, it provides a more accurate measure of arithmetic *fluency* than the arithmetic subtests of the Zareki-R (which are untimed). Norms were based on a sample of 74 8-year-old French children [80]. The Math Fluency subtest of the WJ-II has been shown to have very high reliability ($r_{11}$ = 0.90) [81].

In addition to math skills, children's IQ was estimated using the NEMI-2 standardized intelligence test [82]. The test uses measures of verbal intelligence (i.e., "general knowledge", "vocabulary", and "comparison" subtests) and matrix reasoning (i.e., "Raven's matrices" subtest) to provide a standardized score of full-scale IQ.

**Home numeracy practices.** Home numeracy practices were assessed through an electronic questionnaire given to parents on a tablet. Parents were asked how often they engaged in home learning activities that involved math with their child. To reduce the math-focus of the study, parents were also asked about language or reading activities and activities involving other academic and non-academic domains (e.g., music, biology). The latter, however, were

**Table 2. Ratings associated with frequencies of informal home numeracy practices.**

| Item | Mean (SD) | Min | Max |
|---|---|---|---|
| Weighing / counting shopping[2] | 0.94 (1.03) | 0 | 5 |
| Counting out money[1] | 1.2 (1.05) | 0 | 4 |
| Paying for shopping[2] | 1.06 (1.09) | 0 | 5 |
| Comparing magnitudes[1] | 1 (1.03) | 0 | 4 |
| Playing board games with a numerical dice[1] | 1.29 (0.83) | 0 | 5 |
| Playing number card games[1] | 0.92 (0.82) | 0 | 5 |
| Playing computer / tablet games involving numbers[1] | 0.71 (0.75) | 0 | 3 |
| Making/sorting collections[1] | 0.68 (0.86) | 0 | 4 |
| Measuring lengths/widths[1] | 0.8 (0.7) | 0 | 3 |
| Talking about temperature or speed[2] | 3 (1.54) | 0 | 5 |
| Measuring speeds[1] | 2.65 (1.6) | 0 | 5 |
| Using a calculator[1] | 0.76 (1.02) | 0 | 5 |
| Measuring ingredients while cooking[1] | 1.44 (0.99) | 0 | 5 |
| Talking about the time with a watch or a clock[1] | 3.77 (1.4) | 1 | 5 |
| Talking about the date with a calendar[1] | 2.98 (1.53) | 0 | 5 |
| Dialing phone numbers[2] | 1.06 (1.37) | 0 | 5 |
| Singing songs with numbers[2] | 0.79 (0.86) | 0 | 5 |
| *Average* | *1.47 (1.47)* | | |

All parents (N = 66) were presented with all of the items. Rating: Did not occur/Activity is not relevant to my child was coded 0, Child is doing the activity without parent was coded 1, Parents used to engage in the activity in the past was coded 1, 1–3 times per month was coded 1, Once per week was coded 2, 2–4 times per week was coded 3, Almost daily was coded 4, Daily was coded 5.

[1]Items directly translated from LeFevre et al.'s questionnaire.

[2]Items adapted from LeFevre et al.'s questionnaire to account for the fact that children in the present study are older.

considered filler items and were not analyzed in the present study. Activities were taken from the questionnaire used in LeFevre et al. [20, 22], which is widely used in studies on studies of the HNE (for a review see [25]). Because this questionnaire [20] was designed for parents of children in kindergarten, activities were adapted so that they were appropriate for parents of children in 2nd and 3rd grade (see Tables 2 and 3 for a list of activities and whether they were directly taken or adapted from the original questionnaire). For each activity, parents could choose among 8 response options. If they engaged in the activity with their child at home during the past month, they could indicate the frequency among 5 options: 1–3 times per month (scored 1), Once per week (scored 2), 2–4 times per week (scored 3), Almost daily (scored 4), Daily (scored 5). If they did not engage in the activity with the child within the past month, they had the choice between 3 response options. First, they could indicate that the child did practice the activity within the past month, but without parental involvement. Second, they could indicate that they used to engage in the activity with the child in the past but no longer did at the time of testing (as in [83]). Adding these response options is important because children in the present study were older than in most previous studies on the HNE, which raises the possibility that they might engage in numeracy activity by themselves at home or that some activities in our questionnaires might no longer be relevant [19, 62, 83]. It is thus critical to account for the reasons that might have led parents to not explicitly indicate that they engage in a given activity. In our main analyses, these two response options were each scored 1 because they still point to a somewhat supportive HNE, but we did not want them to weigh excessively on the overall scoring (as our study is focused on present practices that are shared between parents and children). However, we also present supplemental analyses in which these responses are kept separate to disentangle between present activities, past activities, and activities that children are doing alone. Third, the parents could also simply indicate that they did not engage in the activity or that the activity is not relevant. This last option was arguably not indicative of a supportive HNE and was therefore scored 0.

Overall, there were 77 home learning activities mentioned in the questionnaire. In order to make the present study comparable to previous studies on the HNE (see [26]), we used a theoretical rather than data-driven definition of practices [22]. Thirty-six of these activities were math-related, of which 17 were considered *informal* (see Table 2) and 19 were considered *formal* (see Table 3) [20, 38]. Following Skwarchuk et al. [22], formal activities were also considered either *basic* or *advanced*, depending on the complexity of the shared activity. As shown in S1 Table, level of complexity was determined based on the school curriculum (i.e., activities were considered advanced if they were for the most part not practiced in the classroom before Grade 2) Each *advanced* activity was also related to a more *basic* activity. For example, subtracting, multiplying or dividing numbers were considered either *basic* or *advanced* depending on the size of the operands (i.e., single-digit versus double-digit). Similarly, whereas activities such as reading and writing numbers up to 100 were considered relatively *basic*, reading and writing numbers up to 1,000 were considered more *advanced* in 8-year-olds. To disguise the goal of the experiment, the same structure was applied to activities that were not math-related. For instance, S2 and S3 Tables show how literacy activities were also divided into formal versus informal and basic versus advanced.

This design, however, implied that parents had to repeatedly answer questions that were about the same topic but only differed in terms of the complexity of the activity (e.g., "Did you count up to 100 with your child?" followed by "Did you count up to 1,000 with your child?"). Pilot testing revealed that this can make participants feel uncomfortable if they had never engaged in those activities (because they would have had to say so multiple times). To avoid such situations, and because the questionnaire was presented in an electronic format, advanced formal activities (both math and non-math-related) were not presented to parents if

**Table 3. Ratings associated with frequencies of formal home numeracy practices.**

| Skill level | Item | Mean (SD) | Min | Max | N |
|---|---|---|---|---|---|
| **Basic** | **Counting objects**[2] | 1.74 (1.44) | 0 | 5 | 66 |
| | **Counting without objects**[2] | 1.03 (0.94) | 0 | 4 | 66 |
| | **Memorizing results of simple addition problems**[2] | 1.7 (1.38) | 0 | 5 | 66 |
| | **Memorizing multiplication tables**[2] | 2.03 (1.4) | 0 | 5 | 66 |
| | **Comparing quantities**[2] | 1.86 (1.54) | 0 | 5 | 66 |
| | **Adding numbers**[2] | 2.38 (1.3) | 0 | 5 | 66 |
| | **Subtracting single-digit numbers (e.g., 8–1)** [2] | 2.41 (1.51) | 0 | 5 | 66 |
| | **Multiplying single-digit numbers (e.g., 2x3)** [2] | 2.26 (1.34) | 0 | 5 | 66 |
| | **Talking about sharing**[1] | 2.39 (1.81) | 0 | 5 | 66 |
| | **Dividing small numbers (e.g., 6÷2)** [2] | 1.17 (1.29) | 0 | 5 | 66 |
| | **Writing numbers up to 20**[2] | 1.14 (0.92) | 0 | 5 | 66 |
| | **Writing numbers up to 100**[2] | 1.17 (0.98) | 0 | 5 | 66 |
| | **Reading numbers up to 20**[2] | 1 (1.13) | 0 | 5 | 66 |
| | **Reading numbers up to 100**[2] | 1.21 (1.05) | 0 | 5 | 66 |
| | *Average* | *1.68 (1.41)* | | | 66 |
| **Advanced** | **Subtracting double-digit numbers (e.g., 34–16)** [2] | 1.52 (1.42) | 0 | 5 | 61 |
| | **Multiplying double-digit numbers (e.g., 12x6)** [2] | 0.94 (1.09) | 0 | 4 | 59 |
| | **Dividing double-digit numbers (e.g., 12÷4)** [2] | 0.29 (0.62) | 0 | 3 | 39 |
| | **Writing numbers up to 1,000**[2] | 0.89 (0.94) | 0 | 4 | 57 |
| | **Reading numbers up to 1,000**[2] | 1.12 (1.01) | 0 | 5 | 56 |
| | *Average* | *0.95 (1.12)* | | | |

N, number of parents who were presented with the item. Rating: Did not occur/Activity is not relevant to my child was coded 0, Child is doing the activity without parent was coded 1, Parents used to engage in the activity in the past was coded 1, 1–3 times per month was coded 1, Once per week was coded 2, 2–4 times per week was coded 3, Almost daily was coded 4, Daily was coded 5.

[1]Items directly translated from LeFevre et al.'s questionnaire.

[2]Items adapted from the LeFevre et al.'s questionnaire to account for the fact that children in the present study are older.

they stated that the corresponding more basic activity had never occurred at home. In other words, we assumed that parents who had never engaged in a basic activity in the past would be unlikely to now engage in the same activity with a more advanced content (i.e., questions that were not presented were thus considered indicative of activities that parents were not engaged in at present). Table 3 indicates the number of parents who were presented with each item. Frequency ratings associated with informal and formal home numeracy practices are listed on Tables 2 and 3.

**Parent measures.** Parents' math skills as well as attitudes and expectations towards math were assessed using tests and questionnaires. First, arithmetic fluency was assessed using the Math Fluency subtest of the Woodcock-Johnson-III Tests of Achievement (WJ-III) [79]. A standardized score was obtained for each parent using norms from a large sample of adults in the US.

Second, parents completed an electronic questionnaire aimed at evaluating their expectations and attitudes towards math. The questionnaire, which was adapted from LeFevre et al. [20], also included questions about domains other than math (e.g., reading, history, geography). As for the home numeracy practices, this was done to reduce the math focus of the study and these items were not analyzed. After responding to demographic questions and questions about their family (e.g., number of books and games at home, number of hours per day they typically spend with their child, whether the child attended preschool before the age of 3), parents used a five-point rating scale (i.e., Not sure, Strongly disagree, disagree, Agree,

Strongly agree) to describe their current attitudes toward math (e.g. "I find math enjoyable", "I avoid situations that involve math" or "My job involves using math") and their past school experiences ("When I was at school, I was good at math"). These questions were directly taken from the LeFevre et al.'s questionnaire [20] and were simply translated into French. Using a six-point rating scale, they were then asked to provide a subjective estimate of their child's skills in math and other domains (i.e., Not sure, Severe difficulty, Difficulty, Average skills, Good skills, Very good skills). Again, this was directly taken from the LeFevre et al.'s questionnaire [20] and simply translated into French.

Parental expectations regarding math learning for their child were also explored using an adaptation of the LeFevre et al.'s questionnaire [20], in which parents originally rated how important it was for them that their child acquires a given skill by the end of kindergarten. Specifically, we made 3 modifications to the original questionnaire. First, because our questionnaire was presented in an electronic format, parents were forced to give an answer to each item. Therefore, we added a "no opinion" option to the original scale to give them the option to skip an item if they wished to do so. In other words, there were 6 options for each item: Really not important (coded -3), Not important (coded -1), No opinion (coded 0), Important (coded 1), Very important (coded 2), Extremely important (coded 3). Second, parents rated how important it was that their child acquires a given skill by the end of *elementary school* (rather than kindergarten). Third, although we kept all of the original items from the LeFevre et al.'s questionnaire [20] (with the exception of "count to 10" and "count to 20" which were deemed no longer relevant for elementary school children), we added several items that were more appropriate for elementary school children according to the French math curriculum. These are indicated in Table 4. These expectations were considered either *basic* or *advanced*, depending on the corresponding grade level in the French curriculum. Ratings associated with parental expectations are listed on Table 4.

The data and the parental questionnaire can be found on OSF (https://osf.io/vty3c/files/).

**Table 4. Parental expectations regarding math skills to be acquired by children at the end of elementary school.**

| Skill level | Item | Mean (SD) | Min | Max |
|---|---|---|---|---|
| **Basic** | **Count up to 100**[1] | 2.27 (.93) | -1 | 3 |
| | **Count up to 1,000**[1] | 1.94 (1.04) | -1 | 3 |
| | **Read numbers up to 100**[1] | 2.20 (0.89) | -1 | 3 |
| | **Read numbers up to 1,000**[1] | 2.02 (1.02) | -1 | 3 |
| | **Know simple sums without counting on fingers (e.g., 2+2)**[1] | 2.36 (0.75) | -1 | 3 |
| | **Know how to solve complex addition problems (e.g., 15+12)**[1] | 2 (0.78) | 1 | 3 |
| | **Know simple multiplication problems (e.g., 2x6)**[1] | 2.32 (0.70) | 1 | 3 |
| | *Average* | *2.14 (0.75)* | | |
| **Advanced** | **Know complex multiplication problems (e.g., 14x7)**[2] | 1.85 (.96) | -1 | 3 |
| | **Know fractions and how to use them (e.g., 2/3)**[2] | 1.23 (1.16) | -1 | 3 |
| | **Know how to solve division problems (e.g., 30÷5)**[2] | 1.72 (0.98) | -1 | 3 |
| | **Know decimal numbers (e.g., 3.2)**[2] | 1.38 (1.00) | -1 | 3 |
| | **Know how to calculate with decimal numbers**[2] | 1.12 (1.07) | -1 | 3 |
| | **Know probabilities and how to use them**[2] | 0.02 (1.33) | -3 | 2 |
| | *Average* | *1.84 (0.85)* | | |

N = 66; Rating: Really not important was coded -3, Not important was coded -1, No opinion was coded 0, Important was coded 1, Very important was coded 2, Extremely important was coded 3.

[1]Items directly translated from LeFevre et al.'s questionnaire.

[2] Items added for the purpose of our study.

## Data analysis

First, we tested whether math skills increased with the frequency of home numeracy practices (informal, formal basic and formal advanced) using one-tailed bivariate Pearson correlations. We then used scores on math subtests as outcomes of multiple regression analyses that included the predictors frequency of informal, formal basic and formal advanced practices. To test whether any of the relations observed were explained by other measured variables, we adjusted these regressions for children's overall cognitive functioning, parental SES, parental math skills, and time spent with the child at home. Finally, we performed a series of supplemental analyses in which (i) we controlled for the fact that a variable number of participants were presented with advanced items, (ii) we dissociated present practices with parent from present practices without parent and past practices, and (iii) we assessed the discriminant validity of numeracy practices (by investigating literacy practices).

Second, we tested whether home numeracy practices increased with parental traits using one-tailed bivariate Pearson correlations. We then used reported frequency of home numeracy practices as outcomes of multiple regression analyses that included parental traits (parental arithmetic fluency, basic and advanced expectations towards math and math attitudes) as predictors. These regressions were also adjusted for parental SES, parental estimates of their child's math skills time spent with the child at home. Supplemental analyses also dissociated present practices with parent from present practices without parent and past practices. All analyses were conducted in Jamovi version 1.6.3.

## Results

### How do math skills relate to home numeracy practices in early elementary schoolers?

The relations between math skills and home numeracy practices (informal, formal basic and formal advanced) were explored using bivariate Pearson correlations (Table 5) as well as

**Table 5. Pearson correlation coefficients between home numeracy practices and children's math skills (as well as IQ) across all participants.**

|  | 1. | 2. | 3. | 4. | 5. | 6. | 7. | 8. | 9. | 10. |
|---|---|---|---|---|---|---|---|---|---|---|
| *Home numeracy practices* | | | | | | | | | | |
| **1. Informal** | | | | | | | | | | |
| **2. Formal / basic** | 0.46*** | | | | | | | | | |
| **3. Formal / advanced** | 0.28* | 0.76*** | | | | | | | | |
| *Zareki-R* | | | | | | | | | | |
| **4. Quantity estimation** | 0.04 | 0.13 | 0.13 | | | | | | | |
| **5. Symbolic number understanding** | 0.09 | 0.04 | 0.09 | 0.39*** | | | | | | |
| **6. Counting** | 0.03 | -0.18 | -0.14 | 0.28* | 0.31** | | | | | |
| **7. Transcoding** | 0.10 | -0.04 | -0.05 | 0.35** | 0.44*** | 0.40*** | | | | |
| **8. Arithmetic calculation** | 0.16 | 0.02 | 0.17† | 0.35** | 0.46*** | 0.11 | 0.19† | | | |
| *WJ-III* | | | | | | | | | | |
| **9. Arithmetic fluency** | -0.04 | 0.10 | 0.28* | 0.45*** | 0.44*** | 0.16 | 0.16† | 0.53*** | | |
| *Nemi-2* | | | | | | | | | | |
| **10. Full-scale IQ** | 0.07 | <0.01 | 0.03 | 0.33** | 0.40*** | 0.13 | 0.33** | 0.41*** | 0.25* | |

N = 66

***, p < .001

**, p < .01

*, p < .05.

†, p < .1. P values are one-tailed (testing for a positive association).

**Table 6. Effect sizes and t-values associated with multiple regression analyses of numeracy practices on math subtests across all participants.**

| Practice | Quantity Estimation[1] | | Symbolic number understanding[1] | | Counting[1] | | Transcoding[1] | | Arithmetic calculation[1] | | Arithmetic fluency[2] | |
|---|---|---|---|---|---|---|---|---|---|---|---|---|
| | $\eta^2 p$ | t | $\eta^2 p$ | t | $\eta^2 p$ | t | $\eta^2 p$ | t | $\eta^2 p$ | t | $\eta^2 p$ | t |
| **Informal** | <0.001 | -0.115 | 0.010 | 0.773 | 0.016 | 1.008 | 0.016 | 0.992 | 0.045 | 1.715 | 0.003 | -0.461 |
| **Formal basic** | 0.002 | 0.326 | 0.008 | -0.687 | 0.026 | -1.279 | 0.002 | -0.356 | **0.064** | **-2.055** | 0.019 | -1.096 |
| **Formal advanced** | 0.003 | 0.435 | 0.013 | 0.892 | <0.001 | 0.124 | <0.001 | -0.174 | **0.077** | **2.282** | **0.091** | **2.487** |
| **$R^2$** | 0.019 | | 0.021 | | 0.049 | | 0.018 | | 0.105 | | 0.108 | |

N = 66; p < .05 (two-tailed) in bold; η2ps represent effect sizes that can be considered small (0.01), medium (0.06) or large (0.14) [84].

[1]Zareki-R

[2]WJ-III.

multiple regression analyses with numeracy practices as predictors and math subtests as outcomes (Table 6).

As can be seen on Table 5, we only observed a positive correlation between practices that were considered both formal and advanced and the two measures of arithmetic skills: arithmetic calculation and arithmetic fluency. This was confirmed by multiple regression analyses (see Table 6), showing a positive effect of formal advanced numeracy practices on both arithmetic calculation (standardized coefficient or SC = 0.426, 95% confidence interval, or CI = [0.053, 0.799], p = 0.026) and arithmetic fluency (SC = 0.463, 95% CI = [0.091, 0.836], p = 0.016) when controlling for informal and formal basic practices. Critically, these associations remained significant when we added in the models as control variables parental education and income (arithmetic calculation: SC = 0.413, 95% CI = [0.024, 0.803], t = 2.124, p = 0.038, $\eta^2 p$ = 0.070; arithmetic fluency: SC = 0.441, 95% CI = [0.053, 0.829], t = 2.271, p = 0.027, $\eta^2 p$ = 0.079), child's IQ (arithmetic calculation: SC = 0.392, 95% CI = [0.047, 0.737], t = 2.273, p = 0.027, $\eta^2 p$ = 0.078; arithmetic fluency: SC = 0.442, 95% CI = [0.078, 0.806], t = 2.430, p = 0.018, $\eta^2 p$ = 0.088), parental math fluency (arithmetic calculation: SC = 0.428, 95% CI = [0.049, 0.807], t = 2.257, p = 0.028, $\eta^2 p$ = 0.077; arithmetic fluency: SC = 0.490, 95% CI = [0.116, 0.865], t = 2.618, p = 0.011, $\eta^2 p$ = 0.101), and number of hours spent with the child (arithmetic calculation: SC = 0.429, 95% CI = [0.053, 0.805], t = 2.281, p = 0ac.026, $\eta^2 p$ = 0.079; arithmetic fluency: SC = 0.461, 95% CI = [0.085, 0.836], t = 2.453, p = 0.017, $\eta^2 p$ = 0.090). Therefore, home numeracy practices that are formal and relatively challenging are positively associated with children's arithmetic skills, over and above differences in SES, children's overall cognitive functioning, parental math skills, and time spent with the child at home.

We then performed a series of supplemental analyses to control for different factors. First, to minimize parents' discomfort while answering questions about their home practices, we did not present advanced numeracy items to participants who responded that they never engaged in the same activity with a more basic content (see above). This results in a variable number of participants who were presented with each advanced question (and also relies on the assumption that parents who were not presented with a given advanced item never engaged in the practice) (see Table 3). To ensure that this design feature did not affect our findings, we performed a second set of analyses on a subset of participants who were systematically presented with advanced numeracy items (data for this subset of participant are available on https://osf.io/vty3c/files/). To further limit the decrease in sample size associated with such an analysis, we only kept subjects who were presented with the two most frequently seen advanced items (i.e., subtracting and multiplying double-digit numbers) and removed the other 3 advanced

items from the calculation of the frequency of formal advanced home numeracy practices. This resulted in a sample size of 58 participants who were presented with all items. This additional analysis indicated that formal advanced home numeracy practices remained significantly related to both measures of arithmetic skills: arithmetic calculation (SC = 0.482, 95% CI = [0.175, 0.789], p = 0.003) and arithmetic fluency (SC = 0.413, 95% CI = [0.095, 0.731], p = 0.012) (see S4 Table). This was also the case after controlling for parental education and income, child's IQ, parental math fluency, and number of hours spent with the child (arithmetic calculation: all ps < 0.006; arithmetic fluency: all ps < 0.020).

Second, as is standard in studies on the HNE, parents rated the frequency of present numeracy practices with children using a frequency scale. However, because the current study focuses on children who are older than in previous studies on the HNE, we also included two additional response options so that parents had the opportunity to indicate (i) that they used to engage in the activity in the past or (ii) that children engaged in the activity at present but without parental involvement. Although these response options weigh minimally on the calculation of the overall score (see Methods), we ran another set of analyses in which we dissociated present practices with parents from present practices without parents and past practices. S5 Table shows all bivariate relations when practices are dissociated, though these need to be interpreted with caution as all response options were presented concurrently (e.g., it is difficult to interpret a simple relation between skills and present practices with parent as two other response options were also available and not accounted for in such bivariate relations). When present practices with parent, present practices without parent and past practices were all included as separate predictors of multiple regression analyses, there remained a significant positive effect of present formal advanced numeracy practices on arithmetic fluency (SC = 0.484, 95% CI = [0.080, 0.888], p = 0.020) (S6 Table). The positive effect of present formal advanced numeracy practices on arithmetic calculation tended to be significant (SC = 0.392, 95% CI = [-0.020, 0.804], p = 0.061). These associations remained significant for arithmetic fluency (or near significant for arithmetic calculation) when we added in the models as control variables parental education and income (arithmetic fluency: SC = 0.469, 95% CI = [0.050, 0.889], t = 2.245, p = 0.029, $\eta^2 p$ = 0.085; arithmetic calculation: SC = 0.401, 95% CI = [-0.025, 0.828], t = 1.887, p = 0.065, $\eta^2 p$ = 0.062;), child's IQ (arithmetic fluency: SC = 0.481 95% CI = [0.081, 0.881], t = 2.411 p = 0.019, $\eta^2 p$ = 0.096; arithmetic calculation: SC = 0.386, 95% CI = [0.002, 0.771], t = 2.012, p = 0.049, $\eta^2 p$ = 0.069), parental math fluency (arithmetic fluency: SC = 0.510, 95% CI = [0.101, 0.919], t = 2.497, p = 0.016, $\eta^2 p$ = 0.102; arithmetic calculation: SC = 0.393 95% CI = [-0.027, 0.812], t = 1.877, p = 0.066, $\eta^2 p$ = 0.060), and number of hours spent with the child (arithmetic fluency: SC = 0.484, 95% CI = [0.076, 0.892], t = 2.375, p = 0.021, $\eta^2 p$ = 0.093 arithmetic calculation: SC = 0.395, 95% CI = [-0.019, 0.810], t = 1.912, p = 0.061, $\eta^2 p$ = 0.062). Therefore, present practices with parents uniquely contributed to the relation between formal advanced practices and arithmetic skills (particularly arithmetic fluency).

Next, we tested whether the relation between formal advanced practices and arithmetic skills was specific to numeracy activities or could also be obtained with formal advanced literacy practices. In other words, what is the discriminant validity of the math practices items? To answer this question, we analyzed the responses given to the reading practices items (see S2 and S3 Tables for a list of informal and formal literacy practices). We reasoned that these items are comparable to math items in terms of distinction between informal, formal basic and formal advanced practices but that reading activities lack numeracy content. Therefore, if the relation between formal advanced practices and arithmetic skills is specific to the numeracy content, it should not be (or should be more weakly) observed with a literacy content. As can be seen in S7 Table, there was no association between literacy practices and any measures of

math skills (including arithmetic). Therefore, the relation between formal advanced home numeracy practices and arithmetic skills appears to be domain specific. Relatedly, our questionnaire appears to discriminate between different types of activity (even if both are advanced and pertain to academic learning).

### What are the determinants of home numeracy practices in parents of early elementary schoolers?

Finally, the determinants of home numeracy practices were assessed using Pearson correlations (Table 7) and then tested in 3 separate multiple regression analyses with frequencies of informal, formal basic, and formal advanced practices as outcome measures (Table 8).

In each of these analyses, we included as predictors parental arithmetic fluency, parental expectations for math learning (both basic and advanced) and parental attitudes towards math (see Table 8). Overall, multiple regression analyses only revealed positive relations between parental advanced expectations and both formal basic practices (SC = 0.419, 95% CI = [0.125, 0.713], p = 0.006) and formal advanced practices (SC = 0.403, 95% CI = [0.109, 0.696], p = 0.008). These associations remained significant when we added in the models as control variables parental education and income (formal basic practices: SC = 0.377, 95% CI = [0.080, 0.660], t = 2.555, p = 0.013, $\eta^2 p = 0.100$; formal advanced practices: SC = 0.414, 95% CI = [0.100, 0.635], t = 2.751, p = 0.008, $\eta^2 p = 0.114$), parental estimation of children's numerical skills (formal basic practices: SC = 0.420, 95% CI = [0.122, 0.704], t = 2.838, p = 0.006, $\eta^2 p = 0.118$; formal advanced practices: SC = 0.403, 95% CI = [0.095, 0.620], t = 2.720, p = 0.009, $\eta^2 p = 0.110$) and number of hours spent with the child (formal basic practices: SC = 0.431, 95% CI = [0.135, 0.712], t = 2.938, p = 0.005, $\eta^2 p = 0.126$; formal advanced practices: SC = 0.407, 95% CI = [0.098, 0.624], t = 2.749, p = 0.008, $\eta^2 p = 0.112$). Therefore, parents who had the highest expectations for their child in terms of math learning were those who engaged the most frequently in formal home numeracy activities, over and above differences in SES, parental estimates of the child's math skills, and number of hours spent with the child.

**Table 7. Pearson correlation coefficients between home numeracy practices and parental traits across all participants.**

| | 1. | 2. | 3. | 4. | 5. | 6. | 7. | 8. | 9. | 10. | 11. |
|---|---|---|---|---|---|---|---|---|---|---|---|
| *Home numeracy practices* | | | | | | | | | | | |
| 1. Informal | | | | | | | | | | | |
| 2. Formal / basic | 0.28* | | | | | | | | | | |
| 3. Formal / advanced | 0.46*** | 0.76*** | | | | | | | | | |
| *Parental traits* | | | | | | | | | | | |
| 4. Parental education | -0.12 | -0.07 | -0.25 | | | | | | | | |
| 5. Parental income | 0.02 | -0.11 | -0.09 | 0.41*** | | | | | | | |
| 6. Parental basic expectations | 0.17† | 0.12 | 0.03 | 0.09 | 0.11 | | | | | | |
| 7. Parental advanced expectations | 0.23* | 0.35** | 0.28* | -0.10 | 0.03 | 0.56*** | | | | | |
| 8. Parental math attitude | -0.09 | -0.02 | -0.11 | 0.45*** | 0.24* | 0.12 | 0.12 | | | | |
| 9. Parental arithmetic fluency | 0.01 | -0.08 | 0.00 | 0.56*** | 0.37** | 0.17† | -0.03 | 0.42*** | | | |
| 10. Numbers of hours spent with the child | 0.05 | 0.07 | 0.12 | -0.03 | -0.36 | 0.27* | 0.10 | -0.01 | -0.03 | | |
| 11. Parental estimates of the child's math skills | 0.08 | 0.02 | 0.02 | 0.07 | 0.09 | 0.22* | 0.12 | 0.21* | 0.09 | 0.08 | |

N = 66

\*\*\*, p < .001

\*\*, p < .01

\*, p < .05.

†, p < .1. P values are one-tailed (testing for a positive association).

**Table 8. Standardized coefficients, 95% confidence intervals (CI), and effect sizes for the multiple regression analysis of numeracy practices.**

| Predictor | Informal practices | | Formal basic practices | | Formal advanced practices | |
|---|---|---|---|---|---|---|
| | $\eta^2 p$ | t | $\eta^2 p$ | t | $\eta^2 p$ | t |
| Parental arithmetic fluency | 0.005 | 0.527 | 0.013 | 0.899 | 0.001 | -0.257 |
| Parental basic expectations | 0.002 | 0.373 | 0.030 | -1.367 | 0.007 | -0.638 |
| Parental advanced expectations | 0.032 | 1.422 | **0.117** | **2.850** | **0.110** | **2.743** |
| Parental attitudes | 0.019 | -1.099 | 0.030 | -1.367 | 0.002 | -0.357 |
| **R²** | 0.073 | | 0.130 | | 0.132 | |

N = 66; p < .01 (two-tailed) in bold; η2ps represent effect sizes that can be considered small (0.01), medium (0.06) or large (0.14) [84].

We then tested whether the relation between parental advanced expectations and formal practices hold when only present practices with parent are considered (i.e., when ratings associated with present practices without parent and past practices are excluded). S8 Table shows all bivariate relations when practices are dissociated. As can be seen on S9 Table, multiple regression analyses of present practices revealed that positive relations between parental advanced expectations and both formal basic practices (SC = 0.415, 95% CI = [0.121, 0.708], p = 0.006) and formal advanced practices (SC = 0.402, 95% CI = [0.108, 0.696], p = 0.008) remained positive. These associations remained significant when we added in the models as control variables parental education and income (formal basic practices: SC = 0.361, 95% CI = [0.071, 0.650], t = 2.491, p = 0.016, $\eta^2 p$ = 0.095; formal advanced practices: SC = 0.402, 95% CI = [0.099, 0.704], t = 2.656, p = 0.010, $\eta^2 p$ = 0.107), parental estimation of children's numerical skills (formal basic practices: SC = 0.416, 95% CI = [0.120, 0.712], t = 2.810, p = 0.007, $\eta^2 p$ = 0.116; formal advanced practices: SC = 0.403, 95% CI = [0.106, 0.699], t = 2.713, p = 0.009, $\eta^2 p$ = 0.109) and number of hours spent with the child (formal basic practices: SC = 0.427, 95% CI = [0.133, 0.721], t = 2.903, p = 0.005, $\eta^2 p$ = 0.123; formal advanced practices: SC = 0.409, 95% CI = [0.112, 0.705], t = 2.756, p = 0.008, $\eta^2 p$ = 0.112).

## Discussion

In the present study, we took advantage of the extensive behavioral testing of a sample of French 8-year-olds to assess the relation between the HNE and children's mathematical skills in elementary school, while considering the variety of math abilities and types of home numeracy practices. We also aimed to characterize the determinants of a high quality HNE in parents of these children.

### The relation between numeracy practices and math skills in 8-year-olds is restricted to arithmetic skills and practices that are both formal and advanced

Although we tested a wide range of math skills in children, we were only able to find a relation between home numeracy practices and children's arithmetic skills, as measured by both the Zareki-R and the WJ-III tests. To some extent, this finding is consistent with the fact that arithmetic skills are a primary focus of math learning in elementary school and that most of the formal numeracy activities reported by parents of 8-year-olds involve practicing symbolic arithmetic (see Table 3). In fact, previous research in younger children has shown similar correlations between home activities involving arithmetic and children's abilities in basic calculation skills [8, 18, 40, 85, 86]. However, because these studies involved younger children, activities involved more basic arithmetic, such as practicing simple sums. In our study, the positive

relation between formal activities and arithmetic skills was restricted to activities that are not (for the most part) practiced before 2nd grade in school (see S1 Table) and notably involve practicing double-digit arithmetic with children (see Table 3). In other words, the relation between formal math activities and math skills was only observed for activities that were challenging for 8-year-olds. This distinction between basic and advanced formal activities has been suggested before [22] and is broadly supported in the literature [22, 23, 42, 43]. For example, several studies have found a unique positive relation between children's math skills and the frequency of advanced formal numeracy activities [22, 23, 42]. In contrast, the relation between basic numeracy activities and math skills is less clear. For example, the frequency of basic numeracy practices has been found to be either negatively correlated (like in the present study, see Table 6) [42] or not associated with children's math skills [22, 44]. Overall, then, our results confirm that there remains a positive relation between HNE and math skills in elementary school children. However, this relation is specific to practices that must be formal and challenging for children.

### Advanced formal numeracy practices are related to arithmetic skills over and above differences in SES, child's IQ, parent's math skill and number of hours the parent spends at home with the child

An advantage of our study is that we obtained a range of background variables in both children and parents. This makes it possible to dissociate the effect of home numeracy practices from other effects that may also affect math achievement in children. For example, it has long been established that math achievement is associated with family SES [87–89] and with children's overall cognitive functioning [90, 91]. Because these factors are not systematically controlled in studies of the HNE [8, 18, 19, 23, 34, 35, 41, 42], it is not always clear to what extent home numeracy practices specifically relate to math performance. Here, we found that the relation between advanced formal home numeracy practices and children's arithmetic skills remained significant after controlling for SES (i.e., parental income and education) and cognitive functioning (full-scale IQ).

Our analyses also controlled for differences in parent's arithmetic fluency. This is noteworthy because a relation between quality of HNE and academic skills is often interpreted as reflecting a purely environmental influence on children's abilities. However, it is also possible that this association is due (at least in part) to genetic heritability [62, 63]. For example, genetic predispositions to high math or reading achievement from parents may be passed on to children [66]. These parents may also be more likely to maintain high HNE [29, 64, 65]. This is notably suggested by a study from van Bergen and colleagues [63], in which many (but not all) associations between home literacy practices and children's reading fluency were accounted for by parental reading fluency. This suggests that the relation between the HLE and literacy skills may reflect a mixture of environmental and genetic effects. By controlling for differences in arithmetic fluency among parents, our study adds to recent evidence that the relation between the NHE and math skills might not simply reflect a passive gene–environment correlation [29, 62, 67–69]. However, our measure of parental math skills is limited (i.e., it is a 3-minute fluency test) and gathered on one parent only. As such, more research including a more complete assessment of both parents' abilities with a range of measures is needed to completely dissociate effects of heritability from the HNE.

Finally, it is important to consider that the HNE of children in elementary school is likely to be qualitatively different from the HNE of children in preschool. For example, elementary school children are likely to spend less time at home with parents, partly because of increased opportunities for extracurricular involvement and after-school activities. This might also differ from one family to the other. To control for this factor, we included in our analyses the time

spent by the parent with the child at home. Thus, our effect of formal advanced numeracy practices on arithmetic skills may not be (at least completely) accounted for by differences in time spent at home between parent and child. However, two important additional factors in the HNE of elementary school children (as compared to younger children) may deserve further exploration. The first one is related to the introduction of homework. In France, written assignments are fairly limited in elementary school. However, children can still be assigned readings and this may qualitatively affect the HNE. The second one is that, because late elementary schoolers are increasingly involved in extracurricular activities (e.g., clubs, sports) [92, 93], they may be exposed to informal opportunities to practice numerical skills (e.g., keeping track of the score in a basketball game). Given that the present study focused on numeracy activities *at home*, our questionnaire did not specifically test for these opportunities. However, future studies may further explore the potential relation between numeracy practices during extracurricular activities and children's math skills.

## Frequency of formal numeracy practices is related to academic expectations in parents of 8-year-olds

Little is known about the reasons why some parents engage in math activities with their children more than others, though it has been suggested that parental attitudes toward math, beliefs about child math ability and academic expectations may be important factors (at least in parents of younger children) [22, 26, 30, 94, 95]. To date, evidence for an effect of parental attitudes is mixed. While some studies have found that attitudes play a role in the HNE [24, 42, 43, 96, 97], others have failed to find such an effect [20, 22]. However, there is more consistent evidence that parents who have relatively high expectations regarding their child's math achievement also report more frequent numeracy practices with their children [18, 19, 22, 24, 30, 46, 98–100]. For example, both Zippert and Rittle-Johnson and Hart and colleagues [62, 95] have shown that parents' beliefs about their children math abilities are related to the early math environment.

In the present study, we also found a positive relation between frequency of formal practices (both basic and advanced) and parents' math expectations. However, this relation was only observed for expectations that were relatively high. That is, parents of 8-year-olds who reported the most frequent numeracy activities at home were those who had the highest academic expectations for their child in terms of math learning. In contrast, consistent with prior studies on younger children [20, 22], we did not find any relation between parent's math attitude and reported frequency of home numeracy activities. Thus, only high academic expectations appeared to be related to frequency of both basic and advanced home numeracy practices in our sample of 8-year-olds.

## Limitations

To our knowledge, our study is the first to show a specific relation between the HNE and math performance in children from elementary school, while controlling for a wide range of factors and testing a range of math skills. However, it is important to acknowledge several limitations. A first limitation is that our measure of the HNE (as well as our measures of attitudes and expectations) comes from an electronic questionnaire filled out by parents. There are several drawbacks with such questionnaires. First, as in all questionnaires, it is possible that parents may have reported more frequent math practices (and more positive math attitudes and expectations) in an effort to avoid embarrassment and project a more favorable image to the experimenters (i.e., a social desirability bias) [101]. Although we attempted to limit this bias by asking parents to report information about a very wide range of home activities and

expectations (e.g., reading, sport, music), this social desirability bias may still be present. Second, parents may have had difficulty remembering or identifying the math related activities they engaged in at home. This concern may be relatively prevalent in our electronic questionnaire because parents were not able to skip questions (though they often had the option to indicate that they had no opinion or that the question was not relevant). Third, to minimize parents' discomfort, advanced numeracy items were only presented to participants who responded that they had at some point engaged in the same activity with a more basic content. This means that all parents were not presented with all questions. Although these drawbacks may raise concerns about the validity or our questionnaire, it is important to emphasize that the relation we found was quite specific. That is, it was neither found with basic numeracy practices nor with advanced literacy practices. Therefore, our questionnaire appears to be able to discriminate between different related home practices. Nonetheless, it is clear that many of the concerns above could be at least partly alleviated by using direct observations in the lab [40, 85, 102] or at home [85, 103–105] instead of relying on questionnaires. Therefore, our findings would benefit from being replicated by studies using direct observations.

A second limitation is that, although all children were extensively tested on a broader range of measures that what is typically found in the HNE literature, the overall sample size remains limited. Therefore, our results will need to be replicated in larger samples. Nonetheless, given the relative scarcity of data about effects of the HNE on children's math skills outside of the US, we also believe that our data are valuable and may inform future meta-analysis.

A third limitation is that questionnaires were filled by the one parent who was present during the child's testing in the lab. Because this parent was mainly the mother, mothers represent 89% of the parents in our sample. Thus, our measures of the HNE might have been sensibly different had fathers been included in the study. Because most prior studies also predominantly focus on mothers [106], it may be beneficial for future research to gather a more accurate measure of the HNE by presenting surveys to both mothers and fathers.

A fourth limitation is that our questionnaire focused on activities that involved numerical content *per se*. However, math learning has been found to be associated with the development of other non-numerical skills, such as patterning and spatial processing [95, 107]. For example, it has been shown that children's exposure to spatial activities at home is related to arithmetic skills in first grade [108]. Therefore, future studies should assess the effect of home practices beyond numeracy on math skills of older children.

Finally, as is also the case in most previous studies [11, 35, 41, 44, 104], a final limitation is that our data are correlational. In other words, although our results are consistent with the idea that home numeracy practices may foster math skills in children, it is also possible that parents may adapt their activities to their child's interest and skill [109]. To control for this factor, we asked parents to estimate the skill level of their child and included this estimate when assessing the factors predicting the frequency of practices. In other words, there was an effect of high academic expectations on frequency of numeracy practices irrespective of the child's skill (as perceived by the parent). Yet, interventional studies are needed to establish the causality of the relations found here.

## Implications

Our study adds to the growing body of literature indicating that home numeracy experiences may relate to the growth of children's math skills [26, 96]. More specifically, our results suggest that this relation may not only be observed with preschoolers and early elementary schoolers, but also in children who are already learning formal math in elementary school. Importantly, the relation between advanced numeracy practices and math skills in 2nd and 3rd graders can

be considered of medium size in the present study (with a partial η2 between 0.08 and 0.09, see **Table 6**). Even though this estimate is not causal but correlational (and might therefore be overestimated), an effect of this size may have clear educational significance [110, 111]. Specifically, our findings support the idea that children's math achievement might potentially be improved by enhancing the HNE.

We can think of at least 3 ways our findings may inform such interventions. First, our results suggest that such interventions should not be limited to parents of preschoolers, but should also include parents of children in elementary school. Second, our findings indicate that interventions should focus on the content of the numeracy activities, in addition to their frequency. That is, children are more likely to benefit from home numeracy activities that are sufficiently challenging for them (at their grade level or beyond) than from home numeracy activities that they may already be used to. This would require making parents feel capable and helping them identify and plan for opportunities to engage in such relatively advanced activities (for which they may not necessarily feel comfortable at first). This may also require giving parents explicit feedback to maintain their motivation [96]. Third, as in previous studies on younger children [20, 42], we found that expectations of parents towards children's math learning are linked to the quality of the HNE in older children. Therefore, interventions may also benefit from attempting to raise expectations of parents regarding children's math skills. This might involve helping parents overcome math anxiety and misconceptions regarding math (e.g., that some children are 'math persons' and others not; [112]), as well as convincing them of the importance of math skills for the future of their children. This might also require giving them concrete examples of what children in elementary school can learn through daily activities with their parents (in addition to what they learn at school).

## Conclusions

In sum, our study demonstrates that the HNE that parents provide for their children remains related to math skills when children are in elementary school. However, we found that this relation is restricted to activities that are formal and relatively challenging for children. The effect is also specifically present on arithmetic skills, which constitute a main focus of math learning in elementary school. Furthermore, parents who report the most frequent numeracy activities at home are those who tend to have the highest academic expectations for their child. Overall, our study adds to the literature on the HNE by showing that the relation between home numeracy practices and math skills remains present in elementary school children. Although our findings are correlational in nature, they broadly support interventions that may attempt to raise math skills of elementary school children by involving parents and caregivers.

## Supporting information

**S1 Appendix. Subtests of the Zareki-R by math skill.**
(DOCX)

**S1 Table. Formal numeracy activities and approximate grades in which they are mainly practiced in school according to the French math curriculum.**
(DOCX)

**S2 Table. Frequency ratings associated with informal home literacy practices.**
(DOCX)

**S3 Table. Frequency ratings associated with formal home literacy practices.**
(DOCX)

**S4 Table. Effect sizes and t-values associated with multiple regression analyses of numeracy practices on math subtests across the subset of participants who were all presented with the frequency items "subtracting double-digit numbers" and "multiplying double-digit numbers".**
(DOCX)

**S5 Table. Pearson correlation coefficients between present and past home numeracy practices with parent and children's math skills (as well as IQ) across all participants.**
(DOCX)

**S6 Table. Effect sizes and t-values associated with multiple regression analyses of present and past numeracy practices on math subtests across all participants.**
(DOCX)

**S7 Table. Effect sizes and t-values associated with multiple regression analyses of literacy practices on math subtests across all participants.**
(DOCX)

**S8 Table. Pearson correlation coefficients between present and past home numeracy practices with parent and parental traits across all participants.**
(DOCX)

**S9 Table. Standardized coefficients, 95% confidence intervals (CI), and effect sizes for the multiple regression analysis of present numeracy practices with parent.**
(DOCX)

**S1 Data. Data all participants.**
(CSV)

**S2 Data. Data subset participants.**
(CSV)

## Author Contributions

**Conceptualization:** Cléa Girard, Jérôme Prado.

**Data curation:** Cléa Girard.

**Formal analysis:** Cléa Girard, Jérôme Prado.

**Funding acquisition:** Jérôme Prado.

**Methodology:** Cléa Girard, Jérôme Prado.

**Project administration:** Justine Epinat-Duclos, Léa Longo.

**Resources:** Cléa Girard, Thomas Bastelica, Jessica Léone, Justine Epinat-Duclos, Léa Longo.

**Supervision:** Jérôme Prado.

**Writing – original draft:** Cléa Girard.

**Writing – review & editing:** Jérôme Prado.

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
