## [Decision Letter · Decision Letter 0]

5 Feb 2021

PONE-D-20-37746

The relation between home numeracy practices and a variety of math skills in elementary school children

PLOS ONE

Dear Dr. Girard,

Thank you for submitting your manuscript to PLOS ONE. After careful consideration, we feel that it has merit but does not fully meet PLOS ONE’s publication criteria as it currently stands. Therefore, we invite you to submit a revised version of the manuscript that addresses the points raised during the review process.

Both reviewers are very positive about the manuscript, but suggest (rather small) changes to improve the overall quality of the paper. As the reviewers have been very clear in their suggestions for improvement I will not repeat them here. In addition to the suggestions of the reviewers, there are three remaining issues from my own reading of the manuscript that hopefully could be addressed in a revision:

- Using the abbreviation HNE in the Abstract was a bit confusing to me, as it only becomes clear in the main text what it stands for. Could HNE be written in full in the Abstract?

- I would very much prefer it if the correlations could be provided in the main text, as these form the basis of the analyses. 

- Little information is provided about the analyses. Some elaboration on the chosen strategy of analysis would be very helpful. 

We look forward to receiving your revised manuscript.

Kind regards,

Madelon van den Boer

Academic Editor

PLOS ONE

Journal Requirements:

2.Thank you for stating the following in the Funding Section of your manuscript:

"This research was supported by grants from the Agence Nationale de la Recherche (ANR-14-CE30-0002 and ANR-17-CE28-0014) to J.P."

Reviewers' comments:

Reviewer's Responses to Questions

**Comments to the Author**

1. Is the manuscript technically sound, and do the data support the conclusions?

Reviewer #1: Yes

Reviewer #2: Partly

2. Has the statistical analysis been performed appropriately and rigorously? 

Reviewer #1: Yes

Reviewer #2: Yes

3. Have the authors made all data underlying the findings in their manuscript fully available?

Reviewer #1: Yes

Reviewer #2: Yes

4. Is the manuscript presented in an intelligible fashion and written in standard English?

Reviewer #1: No

Reviewer #2: Yes

5. Review Comments to the Author

Reviewer #1: This manuscript further examines the intricate relations between home numeracy practices and children’s numerical skills using a sample of older elementary school children in France. The study is novel in its use of sample and multiple numerical measures. Overall, the strengths of the study include contributing novel questions, data, and results to the existing literature. This manuscript is a nice contribution to the field, however, there are several concerns regarding the clarity of the introduction, hypotheses, and discussion of the study results that should be addressed.

1. The literature review is well-grounded, however, the way in which the literature ties together is unclear. Beginning each paragraph with “second”, “third”, “fourth” does not inherently tie the pieces together. It would be more appropriate to use transition statements across the broad headings in the introduction and clarify why each subheading builds upon the next (if it does).

2. Currently, your headings seem more like place holders rather than formal subheadings. For example can “Relation between the HNE and Math skills in elementary school” become “HNE and Math Skills in Elementary School”?

3. Line 74-76 states studies have found the relation or have failed, but what about studies that have found a significant negative relation (Skwarchuk et al., 2014)?

4. Hypotheses are not stated in the manuscript. It would be helpful if the dense literature review in the introduction was used to inform your hypotheses.

5. The exclusion criteria make sense for neuroimaging methods, but why were these same exclusion criteria included for behavioral, home environment analyses? Make this clearer why you think these exclusions matter for your research questions

6. For participant age, just report mean/min/max rather than single out participants

7. I appreciate your power analysis! Can you cite which tool or calculation you used?

8. Like the authors, I also felt a little uneasy about assuming parents did not do advanced math if they didn’t do basic. However, the authors have noted this as a limitation, and conducted robustness checks to ensure this design did not change findings presented in this manuscript. I really appreciate your thoughtfulness in this and your willingness to provide these robustness checks on OSF. Thank you!!

9. Line 603-605: a paragraph consists of at least 3 sentences. Please consider elaborating on this point in the limitations.

10. The points of limitation are important, however, I also noticed that there was no discussion regarding home practices beyond numeracy (shapes/pattern; see Zippert & Rittle-Johnson, 2020 for review). This may be important to address in the limitations, as not much is known about these topics in older children.

11. In conclusions, “overall, our study adds to the literature on the HNE by showing that the relation between home numeracy practices and math skills is not restricted to the environment of young children” seems as though someone said that it did. I think what you mean here is rather the HNE for older children is also important for math skills, however, that is not what comes across in this last paragraph. Further, the next sentence is a run-on sentence and does not give a strong takeaway message from the interesting results of your study! Please work on re-wording and emphasizing your main results.

Minor

- Line 644 asenxiety should be anxiety

- Line 278 missing the word measure

- In Table 2, include the likert scale meanings in the Notes, same for 3 and 4

Reviewer #2: This was a very interesting study to read. To be honest, I was initially very skeptical of the sample size, and didn’t think I would be able to get past it. But the authors present an idea that I felt very compelling, they were very careful with their conclusions, and were very upfront about the limitations of their sample. After reading the paper, I think that despite the low sample size, this paper will introduce an interesting idea into the literature, that HNE can be important for older kids (something I’ve wondered about, especially why the HNE would be limited to preschool only). Given this, I have only very minor suggestions.

1. Could the authors include a table with the bivariate correlations between all their variables?

2. I think it would be helpful to know how what each questionnaire response option was numerically coded.

3. Can the authors give some explanation for why they coded engaging in an activity before but not now, and never engaged in the activity but the child did as a score of 1? This was something I was left concerned about, especially if 1 was also coded for a different response option. Are those options the same as “did not occur”? I’m not sure!

6. PLOS authors have the option to publish the peer review history of their article (what does this mean?). If published, this will include your full peer review and any attached files.

Reviewer #1: No

Reviewer #2: No

---

## [Author Response · Author response to Decision Letter 0]

9 Mar 2021

Responses to reviews

We wish to thank Dr. van den Boer and the reviewers for their thoughtful and helpful suggestions. Below, we indicate the specific responses to the issues raised and the corresponding changes that we have made in the revised manuscript.

Dr. van den Boer

Both reviewers are very positive about the manuscript, but suggest (rather small) changes to improve the overall quality of the paper. As the reviewers have been very clear in their suggestions for improvement I will not repeat them here. In addition to the suggestions of the reviewers, there are three remaining issues from my own reading of the manuscript that hopefully could be addressed in a revision:

- Using the abbreviation HNE in the Abstract was a bit confusing to me, as it only becomes clear in the main text what it stands for. Could HNE be written in full in the Abstract?

Response: This has now been corrected. 

- I would very much prefer it if the correlations could be provided in the main text, as these form the basis of the analyses. 

Response: We have now added in the main text the Pearson correlation matrices that are at the basis of our analyses (see Table 5 for the correlations between home numeracy practices and math scores and Table 6 for the correlations between home numeracy practices and parental traits).

- Little information is provided about the analyses. Some elaboration on the chosen strategy of analysis would be very helpful. 

Response: We thank the editor for this comment. We have now added a “Data analysis” section in the Method (p. 19). There we state:

“The relations between all math skills and the 3 types of home numeracy practices (informal, formal basic and formal advanced) were first explored using bivariate Pearson correlations and subsequently tested using multiple regression analyses with frequency of informal, formal basic and formal advanced practices as predictors and scores on math subtests as outcomes. To test whether any of the relations observed were explained by other measured variables, we then adjusted these regressions for children’s overall cognitive functioning, parental SES, parental math skills, and time spent with the child at home.

Finally, the relations between home numeracy practices and parental traits were also explored using Pearson correlations and subsequently tested using multiple regression analyses with parental traits (parental arithmetic fluency, basic and advanced expectations towards math and math attitudes) as predictors and reported frequency of the home numeracy practices as outcomes. These regressions were also adjusted for parental SES, parental estimates of their child’s math skills time spent with the child at home. All analyses were conducted in Jamovi version 1.6.3.”

Reviewer 1

(1) The literature review is well-grounded, however, the way in which the literature ties together is unclear. Beginning each paragraph with “second”, “third”, “fourth” does not inherently tie the pieces together. It would be more appropriate to use transition statements across the broad headings in the introduction and clarify why each subheading builds upon the next (if it does).

Response: We are grateful to the reviewer for this comment, which helped us improve the introduction. We have now revised our introduction to make better use of transition statements and clarify the structure (see p. 4 to p. 8).

(2) Currently, your headings seem more like place holders rather than formal subheadings. For example can “Relation between the HNE and Math skills in elementary school” become “HNE and Math Skills in Elementary School”?

Response: We have followed the reviewer’s advice and have changed our headings in the introduction so that they are more formal (see p. 4 to p. 7).

(3) Line 74-76 states studies have found the relation or have failed, but what about studies that have found a significant negative relation (Skwarchuk et al., 2014)?

Response: We thank the reviewer for pointing this out. We now explicitly mention these studies on p. 4. There, we state:

“However, the relation between math skills and the HNE appears to be relatively complex and sometimes inconsistent (for a review, see (25,26). For example, several studies have found a relation between higher quality HNE (i.e., more frequent home numeracy activities) and better numerical skills (e.g., 8,10,19,20,22,24,27–32). However, other studies have failed to find such an association (e.g., 33–36) and some have even found a negative relation (22,37). How can this relative inconsistency be explained?”

(4) Hypotheses are not stated in the manuscript. It would be helpful if the dense literature review in the introduction was used to inform your hypotheses.

Response: Our hypotheses are now more clearly stated at the end of the introduction. On p 8-9, we now state: 

“This extensive data collection allowed us to address two main questions. First, we asked whether there remains a relation between home numeracy practices and math skills in these children, who are older than children typically tested in the HNE literature. Critically, our range of measures allowed us to (a) control for parental math skills, (b) assess which numeracy practices (informal versus formal, basic versus advanced) relate to specific math skills, and (c) evaluate whether the relation would be specific to numeracy practices (i.e., not observed with comparable literacy practices). In keeping with the previous literature on younger children, we expected to observe a relation between the HNE and math skills that would depend upon the type and difficulty of the practice. That is, we anticipated that informal practices would be associated with non-symbolic skills while formal practices would be associated with symbolic skills, especially when those were relatively advanced with respect to children’s age. 

Second, we also aimed to investigate which parental traits (e.g., expectations, attitudes, skills) predict more frequent numeracy practices. Indeed, several studies have investigated the role of parental math attitude and academic expectations in shaping the HNE (see 25,26). Parental attitudes include their level of math anxiety (i.e., their tendency to avoid situations that involve math, 70) but also their use of math in everyday life as well as their evaluation of their own math skills when they were in school (71,72). Academic expectations are typically measured by asking parents how important it is for them that their child acquires a given skill within some time frame (e.g., 22). Overall, although an association between parental attitudes and home numeracy practices remains to be clearly demonstrated (e.g., 20,22), studies have more consistently found a relation between higher academic expectations and increased frequency of numeracy practices at home (18,19,22,30,43). Thus, we expected that the frequency of math activities reported by parents would depend on their academic expectations.”

(5) The exclusion criteria make sense for neuroimaging methods, but why were these same exclusion criteria included for behavioral, home environment analyses? Make this clearer why you think these exclusions matter for your research questions.

Response: We thank the reviewer for this comment and have now specified why we used these exclusion criteria. On p. 9, we state:

“Because studies suggest that the home learning environment may differ in children with low intelligence quotient (IQ) (73) and attention deficit disorder (74), we excluded children who had an IQ lower than the 25th percentile (n=2) and were diagnosed with attention deficit disorder (n=1). Because we also anticipated differences in learning environment of children who experienced and were followed for language difficulties (75), we also excluded children who were seeing a speech-language pathologist on a regular basis (n=3) and were diagnosed with attention deficit disorder (n=1).”

(6) For participant age, just report mean/min/max rather than single out participants

Response: We have followed the reviewer’s advice. On p. 10, we state: 

“Therefore, 66 typically-developing 8-year-olds (M=8.46, min=7.51, max=9.22) with no history of neurological disease, mental disorders of attention deficits were included in the final analyses.”

(7) I appreciate your power analysis! Can you cite which tool or calculation you used?

Response: We thank the reviewer for this comment. We used G*Power (Faul et al., 2007). We added the reference in the manuscript and some more detail about the calculation. On p. 10, we state:

“Considering the smallest estimate of that range (r=0.31), G*Power 3.1 (76) indicated that our final sample (n=66) would provide an achieved power of 84% to detect a similar positive association in a linear regression at α = 0.05 (one tailed).”

(8) Like the authors, I also felt a little uneasy about assuming parents did not do advanced math if they didn’t do basic. However, the authors have noted this as a limitation, and conducted robustness checks to ensure this design did not change findings presented in this manuscript. I really appreciate your thoughtfulness in this and your willingness to provide these robustness checks on OSF. Thank you!!

Response: We thank the reviewer for the comment.

(9) Line 603-605: a paragraph consists of at least 3 sentences. Please consider elaborating on this point in the limitations.

Response: We thank the reviewer for the suggestion. We have now extended this paragraph. It now reads (on p. 29): 

“A second limitation is that, although all children were extensively tested on a broader range of measures that what is typically found in the HNE literature, the overall sample size remains limited. Therefore, our results will need to be replicated in larger samples. Nonetheless, given the relative scarcity of data about effects of the HNE on children’s math skills outside of the US, we also believe that our data are valuable and may inform future meta-analysis.”

(10) The points of limitation are important, however, I also noticed that there was no discussion regarding home practices beyond numeracy (shapes/pattern; see Zippert & Rittle-Johnson, 2020 for review). This may be important to address in the limitations, as not much is known about these topics in older children.

Response: This is good point and we thank the reviewer for bringing it up. We have now added this as a limitation. On p. 30 we state: 

 “A fourth limitation is that our questionnaire focused on activities that involved numerical content per se. However, math learning has been found to be associated with the development of other non-numerical skills, such as patterning and spatial processing (94,106). For example, it has been shown that children’s exposure to spatial activities at home is related to arithmetic skills in first grade (107). Therefore, future studies should assess the effect of home practices beyond numeracy on math skills of older children.”

(11) In conclusions, “overall, our study adds to the literature on the HNE by showing that the relation between home numeracy practices and math skills is not restricted to the environment of young children” seems as though someone said that it did. I think what you mean here is rather the HNE for older children is also important for math skills, however, that is not what comes across in this last paragraph. Further, the next sentence is a run-on sentence and does not give a strong takeaway message from the interesting results of your study! Please work on re-wording and emphasizing your main results.

Response: We have followed the reviewer’s advice and have reworded the conclusion. We also end on a sentence that emphasizes an important implication of our study. On p. 31-32, this revised section reads: 

“Overall, our study adds to the literature on the HNE by showing that the relation between home numeracy practices and math skills remains present in elementary school children. Although our findings are correlational in nature, they broadly support interventions that may attempt to raise math skills of elementary school children by involving parents and caregivers.”

- Line 644 asenxiety should be anxiety

Response: This has been corrected

- Line 278 missing the word measure

Response: This has been corrected.

- In Table 2, include the likert scale meanings in the Notes, same for 3 and 4

Response: Likert scale meanings have been added in Tables 2, 3 and 4.

Reviewer 2

(1) Could the authors include a table with the bivariate correlations between all their variables?

Response: We have now added in the main text the Pearson correlation matrices that are at the basis of our analyses (see Table 5 for the correlations between home numeracy practices and math scores and Table 6 for the correlations between home numeracy practices and parental traits).

(2) I think it would be helpful to know how what each questionnaire response option was numerically coded.

Response: We have now added Likert scale meanings in Tables 2, 3 and 4. Additionally, we now provide more details in the main text. On p. 13, we state for the home numeracy questionnaire:

“Following LeFevre et al. (20), a six-point rating scale was used: Did not occur/Activity is not relevant to my child (coded 0), 1-3 times per month (coded 1), Once per week (coded 2), 2-4 times per week (coded 3), Almost daily (coded 4), Daily (coded 5). Given that children in the present study were older than children in LeFevre et al. (20), we added two other response options. First, if parents used to engage in a given activity in the past but no longer did at the time of testing, they could say so. Second, if parents had never engaged in the activity with the child but the child was doing that activity on her own, they could also say so. Each of these options was coded with the lowest non-0 rating (i.e., 1), as they might indicate a somewhat supportive home numeracy environment but do not convey any specific information about frequency (they therefore have minimal weigh on the overall final rating).”

On p. 17, we also state for the academic expectation questionnaire:

“First, because our questionnaire was presented in an electronic format, we added a “no opinion” option to the original scale, such that it had 6 options: Really not important (coded -3), Not important (coded -1), No opinion (coded 0), Important (coded 1), Very important (coded 2), Extremely important (coded 3).”

(3) Can the authors give some explanation for why they coded engaging in an activity before but not now, and never engaged in the activity but the child did as a score of 1? This was something I was left concerned about, especially if 1 was also coded for a different response option. Are those options the same as “did not occur”? I’m not sure!

Response: We thank the reviewer for this comment. These options are coded as 1 and not as “did not occur” (which is coded as 0) as they still indicate a somewhat supportive home environment. We now explain more clearly the rationale for this decision on p. 13. There, we state:

“Given that children in the present study were older than children in LeFevre et al. (20), we added two other response options. First, if parents used to engage in a given activity in the past but no longer did at the time of testing, they could say so. Second, if parents had never engaged in the activity with the child but the child was doing that activity on her own, they could also say so. Each of these options was coded with the lowest non-0 rating (i.e., 1), as they might indicate a somewhat supportive home numeracy environment but do not convey any specific information about frequency (they therefore have minimal weigh on the overall final rating).”

---

## [Decision Letter · Decision Letter 1]

9 May 2021

PONE-D-20-37746R1

The relation between home numeracy practices and a variety of math skills in elementary school children

PLOS ONE

Dear Dr. Girard,

Thank you for submitting your manuscript to PLOS ONE. After careful consideration, we feel that it has merit but does not fully meet PLOS ONE’s publication criteria as it currently stands. Therefore, we invite you to submit a revised version of the manuscript that addresses the points raised during the review process.

Both reviewers on the original manuscript and myself have reviewed the revision of your manuscript. Reviewer 1 was pleased with the changes made and had no further comments. Reviewer 2, however, raised one main issue; a request for an additional analysis, required to better understand current findings and increase comparability to previous studies. From my own reading of the manuscript there are two minor points: on page 9 (lines 199 and 202) a participant with ‘attention deficit disorder’ seems to be mentioned twice and on page 17 (lines 344/345) it is not clear to me how the added answer options are related to the fact that the questionnaire was administered in an electronic format… maybe the sentence could be reworded? I would like to invite you to address these few points in an additional minor revision.

We look forward to receiving your revised manuscript.

Kind regards,

Madelon van den Boer

Academic Editor

PLOS ONE

Journal Requirements:

Reviewers' comments:

Reviewer's Responses to Questions

**Comments to the Author**

1. If the authors have adequately addressed your comments raised in a previous round of review and you feel that this manuscript is now acceptable for publication, you may indicate that here to bypass the “Comments to the Author” section, enter your conflict of interest statement in the “Confidential to Editor” section, and submit your "Accept" recommendation.

Reviewer #1: All comments have been addressed

Reviewer #2: (No Response)

2. Is the manuscript technically sound, and do the data support the conclusions?

Reviewer #1: Yes

Reviewer #2: (No Response)

3. Has the statistical analysis been performed appropriately and rigorously? 

Reviewer #1: Yes

Reviewer #2: (No Response)

4. Have the authors made all data underlying the findings in their manuscript fully available?

Reviewer #1: Yes

Reviewer #2: (No Response)

5. Is the manuscript presented in an intelligible fashion and written in standard English?

Reviewer #1: Yes

Reviewer #2: (No Response)

6. Review Comments to the Author

Reviewer #1: (No Response)

Reviewer #2: Now that I know how the variables were coded, and see the rationale for the recoding of the HNE frequency questionnaire, I am more skeptical about the idea of the extra two responses options for the HNE (engaging in an activity before but not now, and never engaged in the activity but the child did) being coded as 1, the same as a low frequency option. To me this doesn’t seem to be a valid choice given the scale is a frequency scale, and also this choice means this work cannot be compared to other HNE work. I personally think these options should not be used at all, but given the authors likely disagree, I believe that at the very least they should do a sensitivity analysis to see if it matters. I would like to know if they recoded those response to missing, what their correlation and regression results would look like (and I think they should be provided somewhere for future meta-analysists).

7. PLOS authors have the option to publish the peer review history of their article (what does this mean?). If published, this will include your full peer review and any attached files.

Reviewer #1: No

Reviewer #2: No

---

## [Author Response · Author response to Decision Letter 1]

8 Jun 2021

Responses to reviews

We wish to thank Dr. van den Boer and reviewer #2 for their further comments on the manuscript. Below, we indicate the specific responses to the issues raised and the corresponding changes that we have made in the revised manuscript.

Dr. van den Boer

on page 9 (lines 199 and 202) a participant with ‘attention deficit disorder’ seems to be mentioned twice 

Response: We thank the editor for spotting this inconsistency. This was indeed an error. Only one participant was excluded for attention deficit disorder. This has been corrected in the new version of the manuscript. On p. 8/9, this section now reads:

“Because studies suggest that the home learning environment may differ in children with low intelligence quotient (IQ) (73) and attention deficit disorder (74), we excluded children who had an IQ lower than the 25th percentile (n=2) and were diagnosed with attention deficit disorder (n=1). Because we also anticipated differences in the learning environment of children who experienced and were followed for language difficulties (75), we also excluded children who were seeing a speech-language pathologist on a regular basis (n=3) and had a delay in speech and language acquisition (n=1).”

on page 17 (lines 344/345) it is not clear to me how the added answer options are related to the fact that the questionnaire was administered in an electronic format… maybe the sentence could be reworded?

Response: What we meant here is that because the questionnaire was in an electronic format, participants were forced to provide an answer to each item. Adding a ‘no opinion’ option gave them the opportunity to skip items if they wished to do so. This has now been clarified in the manuscript. On p. 16, we state:

“First, because our questionnaire was presented in an electronic format, parents were forced to give an answer to each item. Therefore, we added a “no opinion” option to the original scale to give them the option to skip an item if they wished to do so. In other words, there were 6 options for each item: Really not important (coded -3), Not important (coded -1), No opinion (coded 0), Important (coded 1), Very important (coded 2), Extremely important (coded 3).” 

Reviewer 2

Now that I know how the variables were coded, and see the rationale for the recoding of the HNE frequency questionnaire, I am more skeptical about the idea of the extra two responses options for the HNE (engaging in an activity before but not now, and never engaged in the activity but the child did) being coded as 1, the same as a low frequency option. To me this doesn’t seem to be a valid choice given the scale is a frequency scale, and also this choice means this work cannot be compared to other HNE work. I personally think these options should not be used at all, but given the authors likely disagree, I believe that at the very least they should do a sensitivity analysis to see if it matters. I would like to know if they recoded those response to missing, what their correlation and regression results would look like (and I think they should be provided somewhere for future meta-analysists).

Response: We acknowledge that these two responses are typically not added to measures of the HNE in other studies. This is most certainly because the large majority of previous studies to date have focused on younger children who are less likely to either engage in numeracy activities by themselves or for whom most activities remain relevant. Given that children in our sample are 8 years old, it was critical for us to ask parents why they did not state that an activity was practiced during the past month. This choice was notably motivated by Lefevre et al. (LeFevre et al., 2010) who, when discussing discrepancy with an earlier study involving school-age children (LeFevre et al., 2009), observed that “basic skills were probably already mastered by many of the children in LeFevre et al. (who were in Kindergarten, Grade 1 and Grade 2) and thus were no longer relevant to predicting their more advanced numeracy skills.” Similarly, school-age children are more likely to engage in numeracy activities by themselves than younger children (e.g., because of homework from school), so it is important to gather this information as well. 

In our opinion, scoring these response options as ‘1’ (the lowest non-0 option) makes the most sense because (i) we do not think any of these options are equivalent to the option “I did not engage in the activity” (they clearly indicate a somewhat more supporting HNE), but at the same time (ii) we do not want them to weigh excessively on the overall scoring (as our study is focused on present practices that are shared between parents and children). On p. 12, we now more clearly explain our rationale:

“For each activity, parents could choose among 8 response options. If they engaged in the activity with their child at home during the past month, they could indicate the frequency among 5 options: 1-3 times per month (scored 1), Once per week (scored 2), 2-4 times per week (scored 3), Almost daily (scored 4), Daily (scored 5). If they did not engage in the activity with the child within the past month, they had the choice between 3 response options. First, they could indicate that the child did practice the activity within the past month, but without parental involvement (as in 83). Second, they could indicate that they used to engage in the activity with the child in the past but no longer did at the time of testing. Adding these response options is important because children in the present study were older than in most previous studies on the HNE, which raises the possibility that they might engage in numeracy activity by themselves at home or that some activities in our questionnaires might no longer be relevant (19,62,83). It is thus critical to account for the reasons that might have led parents to not explicitly indicate that they engage in a given activity. In our main analyses, these two response options were each scored 1 because they still point to a somewhat supportive HNE, but we did not want them to weigh excessively on the overall scoring (as our study is focused on present practices that are shared between parents and children). However, we also present supplemental analyses in which these responses are kept separate to disentangle between present activities, past activities, and activities that children are doing alone. Third, the parents could also simply indicate that they did not engage in the activity or that the activity is not relevant. This last option was arguably not indicative of a supportive HNE and was therefore scored 0.”

That being said, we appreciate the reviewer’s point about being able to compare our results to other studies and we thank the reviewer for the suggestion. We have followed the advice and now present additional results in which present activities are dissociated from past activities and activities that children are doing alone. It is important, however, to consider that these two response options (i.e., past activities and activities that children are doing alone) remained presented to parents alongside the frequency scale of shared present activities. In other words, the mere presence of these options among the possible responses is likely to have influenced parents’ ratings and we cannot simply ignore this factor. Thus, we systematically added responses to past activities and responses to activities that children are doing alone as predictors of no interest in regression analyses investigating the effect of present activity shared with parents on math skills (see p. 18 for a description of the new analyses). We also added tables showing bivariate relations in supplementary materials, but these need to be interpreted with caution because the relation between present activities shared with parents and math skills is not controlled for the fact that other response options were presented. 

On p. 22/23, we present the results of the new analyses in which present activities are dissociated from past activities and activities children were doing alone. As can be seen from the results, we still observed a significant relation between present activities and arithmetic fluency (there was only a trend for arithmetic calculation). This new paragraph reads:

“Second, as is standard in studies on the HNE, parents rated the frequency of present numeracy practices with children using a frequency scale. However, because the current study focuses on children who are older than in previous studies on the HNE, we also included two additional response options so that parents had the opportunity to indicate (i) that they used to engage in the activity in the past or (ii) that children engaged in the activity at present but without parental involvement. Although these response options weigh minimally on the calculation of the overall score (see Methods), we ran another set of analyses in which we dissociated present practices with parents from present practices without parents and past practices. S5 Table shows all bivariate relations when practices are dissociated, though these need to be interpreted with caution as all response options were presented concurrently (e.g., it is difficult to interpret a simple relation between skills and present practices with parent as two other response options were also available and not accounted for in such bivariate relations). When present practices with parent, present practices without parent and past practices were all included as separate predictors of multiple regression analyses, there remained a significant positive effect of present formal advanced numeracy practices on arithmetic fluency (SC = 0.484, 95% CI = [0.080, 0.888], p = 0.020) (S6 Table). The positive effect of present formal advanced numeracy practices on arithmetic calculation tended to be significant (SC = 0.392, 95% CI = [-0.020, 0.804], p = 0.061). These associations remained significant for arithmetic fluency (or near significant for arithmetic calculation) when we added in the models as control variables parental education and income (arithmetic fluency: SC = 0.469, 95% CI = [0.050, 0.889], t = 2.245, p = 0.029, η²p = 0.085; arithmetic calculation: SC = 0.401, 95% CI = [-0.025, 0.828], t = 1.887, p = 0.065, η²p =0.062;), child’s IQ (arithmetic fluency: SC = 0.481 95% CI = [0.081, 0.881], t = 2.411 p = 0.019, η²p = 0.096; arithmetic calculation: SC = 0.386, 95% CI = [0.002, 0.771], t = 2.012, p = 0.049, η²p = 0.069), parental math fluency (arithmetic fluency: SC = 0.510, 95% CI = [0.101, 0.919], t = 2.497, p = 0.016, η²p = 0.102; arithmetic calculation: SC = 0.393 95% CI = [-0.027, 0.812], t = 1.877, p = 0.066, η²p = 0.060), and number of hours spent with the child (arithmetic fluency: SC = 0.484, 95% CI = [0.076, 0.892], t = 2.375, p = 0.021, η²p = 0.093 arithmetic calculation: SC = 0.395, 95% CI = [-0.019, 0.810], t = 1.912, p = 0.061, η²p = 0.062). Therefore, present practices with parents uniquely contributed to the relation between formal advanced practices and arithmetic skills (particularly arithmetic fluency).”

On p. 26, we now also show that there remains a significant relation between parental advanced expectations and present practices (when past activities and activities that children are doing alone are excluded from the scoring). There we state:

“We then tested whether the relation between parental advanced expectations and formal practices hold when only present practices with parent are considered (i.e., when ratings associated with present practices without parent and past practices are excluded). S8 Table shows all bivariate relations when practices are dissociated. As can be seen on S9 Table, multiple regression analyses of present practices revealed that positive relations between parental advanced expectations and both formal basic practices (SC = 0.415, 95% CI = [0.121, 0.708], p = 0.006) and formal advanced practices (SC = 0.402, 95% CI = [0.108, 0.696], p = 0.008) remained positive. These associations remained significant when we added in the models as control variables parental education and income (formal basic practices: SC = 0.361, 95% CI = [0.071, 0.650], t = 2.491, p = 0.016, η²p = 0.095; formal advanced practices: SC = 0.402, 95% CI = [0.099, 0.704], t = 2.656, p = 0.010, η²p = 0.107), parental estimation of children’s numerical skills (formal basic practices: SC = 0.416, 95% CI = [0.120, 0.712], t = 2.810, p = 0.007, η²p =0.116; formal advanced practices: SC = 0.403, 95% CI = [0.106, 0.699], t = 2.713, p = 0.009, η²p = 0.109) and number of hours spent with the child (formal basic practices: SC = 0.427, 95% CI = [0.133, 0.721], t = 2.903, p = 0.005, η²p =0.123; formal advanced practices: SC = 0.409, 95% CI = [0.112, 0.705], t = 2.756, p = 0.008, η²p = 0.112).”

We thank the reviewer for their valuable feedback, and we hope that the new analyses will help other researchers use our results in future meta-analyses.

---

## [Decision Letter · Decision Letter 2]

16 Jul 2021

The relation between home numeracy practices and a variety of math skills in elementary school children

PONE-D-20-37746R2

Dear Dr. Girard,

We’re pleased to inform you that your manuscript has been judged scientifically suitable for publication and will be formally accepted for publication once it meets all outstanding technical requirements.

Kind regards,

Madelon van den Boer

Academic Editor

PLOS ONE

Additional Editor Comments (optional):

Thank you for your careful consideration of the latest comments of the reviewer and myself. We both have no further comments and believe the manuscript can be accepted for publication.

Reviewers' comments:

Reviewer's Responses to Questions

**Comments to the Author**

1. If the authors have adequately addressed your comments raised in a previous round of review and you feel that this manuscript is now acceptable for publication, you may indicate that here to bypass the “Comments to the Author” section, enter your conflict of interest statement in the “Confidential to Editor” section, and submit your "Accept" recommendation.

Reviewer #2: (No Response)

2. Is the manuscript technically sound, and do the data support the conclusions?

Reviewer #2: (No Response)

3. Has the statistical analysis been performed appropriately and rigorously? 

Reviewer #2: (No Response)

4. Have the authors made all data underlying the findings in their manuscript fully available?

Reviewer #2: (No Response)

5. Is the manuscript presented in an intelligible fashion and written in standard English?

Reviewer #2: (No Response)

6. Review Comments to the Author

Reviewer #2: (No Response)

7. PLOS authors have the option to publish the peer review history of their article (what does this mean?). If published, this will include your full peer review and any attached files.

Reviewer #2: No

---

## [Editor Report · Acceptance letter]

9 Sep 2021

PONE-D-20-37746R2 

The relation between home numeracy practices and a variety of math skills in elementary school children 

Dear Dr. Girard:

I'm pleased to inform you that your manuscript has been deemed suitable for publication in PLOS ONE. Congratulations! Your manuscript is now with our production department. 

Kind regards, 

on behalf of

Dr. Madelon van den Boer 

Academic Editor

PLOS ONE